# De Novo p.Asp3368Gly Variant of Dystrophin Gene Associated with X-Linked Dilated Cardiomyopathy and Skeletal Myopathy: Clinical Features and In Silico Analysis

**DOI:** 10.3390/ijms25052787

**Published:** 2024-02-28

**Authors:** Maria d’Apolito, Alessandra Ranaldi, Francesco Santoro, Sara Cannito, Matteo Gravina, Rosa Santacroce, Ilaria Ragnatela, Alessandra Margaglione, Giovanna D’Andrea, Grazia Casavecchia, Natale Daniele Brunetti, Maurizio Margaglione

**Affiliations:** 1Medical Genetics, Department of Clinical and Experimental Medicine, University of Foggia, 71122 Foggia, Italy; maria.dapolito@unifg.it (M.d.); alessandra.ranaldi@unifg.it (A.R.); sara.cannito@unifg.it (S.C.); rosa.santacroce@unifg.it (R.S.); giovanna.dandrea@unifg.it (G.D.); 2Department of Medical and Surgical Sciences, University of Foggia, 71122 Foggia, Italy; dr.francesco.santoro.it@gmail.com (F.S.); matteo.gravina@unifg.it (M.G.); ilaria.ragnatela@unifg.it (I.R.); natale.brunetti@unifg.it (N.D.B.); 3Cardiology Unit, University Polyclinic Hospital of Foggia, 71122 Foggia, Italy

**Keywords:** DMD, dystrophin (DMD), dilated cardiomyopathy (DCM), skeletal muscle, dystrophy

## Abstract

Dystrophin (*DMD*) gene mutations are associated with skeletal muscle diseases such as Duchenne and Becker Muscular Dystrophy (BMD) and X-linked dilated cardiomyopathy (XL-DCM). To investigate the molecular basis of DCM in a 37-year-old woman. Clinical and genetic investigations were performed. Genetic testing was performed with whole exome sequencing (WES) using the Illumina platform. According to the standard protocol, a variant found by WES was confirmed in all available members of the family by bi-directional capillary Sanger resequencing. The effect of the variant was investigated by using an in silico prediction of pathogenicity. The index case was a 37-year-old woman diagnosed with DCM at the age of 33. A germline heterozygous A>G transversion at nucleotide 10103 in the *DMD* gene, leading to an aspartic acid–glycine substitution at the amino acid 3368 of the DMD protein (c.10103A>G p.Asp3368Gly), was identified and confirmed by PCR-based Sanger sequencing of the exon 70. In silico prediction suggests that this variant could have a deleterious impact on protein structure and functionality (CADD = 30). The genetic analysis was extended to the first-degree relatives of the proband (mother, father, and sister) and because of the absence of the variant in both parents, the p.Asp3368Gly substitution was considered as occurring de novo. Then, the direct sequencing analysis of her 8-year-old son identified as hemizygous for the same variant. The young patient did not present any signs or symptoms attributable to DCM, but reported asthenia and presented with bilateral calf hypertrophy at clinical examination. Laboratory testing revealed increased levels of creatinine kinase (maximum value of 19,000 IU/L). We report an early presentation of dilated cardiomyopathy in a 33-year-old woman due to a de novo pathogenic variant of the dystrophin (*DMD*) gene (p.Asp3368Gly). Genetic identification of this variant allowed an early diagnosis of a skeletal muscle disease in her son.

## 1. Introduction

Dilated cardiomyopathy (DCM) is featured by left or biventricular dilatation and systolic dysfunction in the absence of secondary causes such as coronary artery disease [1]. DCM is a common cause of heart failure, occurring with a prevalence of 1 in 250 in the general population, and the leading indication for cardiac transplantation [2]. DCM is correlated with genetic and non-genetic causes, with approximately 40% of DCM cases presenting a genetic cause. DCM is considered to be familial when more than one first-degree relative has been diagnosed with DCM or has had sudden cardiac death at a young age [3,4]. Clinical manifestations have been associated with rare variants in at least 45 genes that can be grouped into four major categories: proteins forming the myocyte cytoskeleton, sarcomeric proteins, nuclear envelope proteins, and calcium homeostasis and mitochondrial function regulators [5,6]. Causative genes of DCM can be distinguished in different categories, with a proven (10 genes: *TTN*, *FLNC*, *LMNA*, *BAG3*, *MYH7*, *RBM20*, *TNNT2*, *DES*, *SCN5A*, and *TNNC1*), strong (*DSP*), and moderate (6 genes: *TPM1*, *TNNI3*, *ACTC1*, *VCL*, *NEXN*, and *JPH2*) association with DCM [7]. The most common genetic causes of DCM are pathogenic variants within the titin (*TTN*) gene [8]. Truncating *TTN* variants (TTNtv) are associated with 25% of familial DCM cases and 18% of sporadic DCM cases. *LMNA* is another important gene strongly associated with DCM [9] Pathogenic *LMNA* variants cause debilitating bradyarrhythmia (high grade heart blocks) and tachyarrhythmia (atrial fibrillation or ventricular tachycardia) [9]. In patients carrying causative *LMNA* variants, therapies are focused on preventing sudden death and heart failure progression with the implantation of ICDs. Approximately 10% of cases presenting with genetic DCM carry damaging variants in genes encoding for desmosomes. Desmosomal genes are correlated with higher rates of malignant ventricular arrhythmia with one of them being known as the *DES* gene, which encodes for the muscle-specific protein desmin. Most of the patients with a *DES* variant present with a combination of skeletal and cardiac myopathy, and only 22% of them have an isolated cardiac phenotype [10]. Most familial DCM has an autosomal dominant inheritance pattern; however, other inheritance patterns have been identified including autosomal recessive, X-linked, mitochondrial, and polygenic [11]. X-linked dilated cardiomyopathy (XLDCM) is caused by pathogenic variants occurring within the dystrophin (*DMD*) gene and is often phenotypically indistinguishable from DCM due to other causes. Usually, *DMD* variants are associated with Duchenne and Becker Muscular Dystrophy (BMD), which are characterized by skeletal muscle weakness or limb girdle dystrophy. In addition, variants in the *DMD* gene cause a severe form of dilated cardiomyopathy associated with high rates of heart failure, heart transplantation, and ventricular arrhythmias. XLDCM was first described by Berko and Swift in 1987 as a rapidly progressive form of DCM [12]. XLDCM is an extremely rare and distinct phenotype of dystrophinopathy (<3%) characterized by preferential cardiac involvement without any overt skeletal myopathy [13]. Patients with isolated XLDCM may have complete dystrophin loss in cardiac muscle. Disease manifestations of *DMD*-associated DCM may not be restricted to young males with DCM and raised creatine kinases (CK) levels, but also involve females that could have mild DCM later in life [14]. Several patients have been reported with elevated CK levels without associated muscle weakness and causative mutation in the DMD gene [15]. Patients with Becker muscular dystrophy have a milder phenotype of skeletal muscle involvement and sometimes present with DMC. XLDCM with *DMD* variants or mild Becker muscular dystrophy with DCM are considered to belong to the same clinical spectrum. The relationship between *DMD* gene variants and cardiomyopathy remains unclear. Results from the analyses of XLDCM cases prompted some hypothetical mechanisms that correlate different isoforms of dystrophin with certain reported variants [16].

We report the case of a 37-year-old woman diagnosed with DCM at the age of 33, who underwent genetic testing with whole exome sequencing (WES) using the Illumina platform. The identified gene variant was investigated by using an in silico prediction of pathogenicity and within-family linkage analysis.

## 2. Case Presentation

The index patient was a 37-year-old Italian woman diagnosed at the age of 33 years with DCM and slight left ventricular systolic dysfunction.

All family members available for assessment underwent a detailed medical cardiological examination. Clinical and genetic investigations were performed in accordance with the Helsinki declaration and based on written informed consent for clinical and genetic testing. Written informed consent was requested from their legal representatives for subjects under 18 years of age. All data presented in the manuscript were properly anonymized. The study was approved by the local ethics committee (protocol code 3261/CE/20).

We performed exome sequencing and analysis as previously described [17]. In brief, DNA was isolated from venous blood samples using the Automated Extraction Instruments, MagCore^®^ Plus II (Diatech Pharmacogenetics, Iesi, Italy) according to manufacturer instructions. Enrichment was performed with Illumina DNA prep with Enrichment (Illumina, San Diego, CA, USA) to capture all coding regions and exon–intron junctions (±50 bps) followed by paired-end sequencing using the Illumina NextSeq 550 instrument (Illumina, San Diego, CA, USA). The raw data were then processed according to the Genome Analysis Toolkit (GATK 1.6). The reads were aligned to the human genome (GRCh37) and variant calling was analysed using the software BaseSpace Variant Interpreter Version 2.17.0.60 (Illumina, San Diego, CA, USA) [18]. Variants with a heterozygous allele frequency >1%, a homozygous allele frequency >0.01% (GnomAD heterozygous and homozygous frequency of all populations; https://gnomad.broadinsitute.org/), (accessed on 1 January 2024) and a combined annotation-dependent depletion (CADD) score ≤20 were discarded. 

The gene list of DCM studied is provided in the Appendix A. The gene list compiles dilated cardiomyopathy-associated candidate genes, as well as dilated cardiomyopathy genes candidate in animal models.

Segregation analysis was performed using Sanger sequencing according to standard methods [18]. In brief, the region of interest was amplified by PCR with *specific* 20-base *primers* (DMD_forward 5′-AAGTGTCATGGGGCAGAAGA-3′; DMD_reverse 5′-CACGTTTCCATGTTGTCCCC-3′). Cycle sequencing was performed on the PCR products using BigDye™ Terminator V1.3 (Thermo Fisher Scientific, Waltham, MA, USA), followed by BigDye XTerminator™ Purification (Applied Biosystems, Waltham, MA, USA), and sequencing was performed using SeqStudio (Thermo Fisher Scientific, Waltham, MA, USA) capillary sequencing [19].

To further prioritize the candidate gene variants, functional annotation was undertaken based on the effect on protein function and a priori knowledge of the phenotype. In order to define the severity of a sequence variation and the likelihood of its impact on protein functionality, in silico pathogenicity prediction was performed using different bioinformatic tools for nonsynonymous amino acid substitutions and for splicing variants. The NCBI GenBank accession number NM_004006.3 and UniProt identifier P11532 were used as reference sequences. The effect of the p.Asp3368Gly substitution was explored by using in silico pathogenicity prediction tools (PolyPhen-2, http://genetics.bwh.harvard.edu/pph2/; SIFT, http://sift.jcvi.org/siftbin/retrieve_enst.pl; REVEL and MetaLR https://sites.google.com/site/jpopgen/dbNSFP). The CADD (combined annotation-dependent depletion) tool (http://cadd.gs.washington.edu/home) was used to integrate multiple annotations for the deleteriousness of single nucleotide variants (accessed on 1 January 2024).

### 2.1. Proband Clinical Characteristics

The proband was a 37-year-old Italian woman diagnosed with DCM at the age of 33 years, when she was admitted to the Emergency Department due to pulmonary edema with evidence of left ventricular dilatation and a severely reduced ejection fraction shown with echocardiography. Coronary arteries were free from hemodynamically significant lesions shown with coronary angiography. Cardiac magnetic resonance imaging (MRI) showed severe left and right ventricular systolic dysfunction (LVEF = 27%, RVEF = 34%) and the presence of late gadolinium enhancement (LGE) in the mesocardial inferior interventricular septum, infero-lateral wall of the left ventricle, and free wall of the right ventricle (Figure 1a–d).

In the same year, she underwent a subcutaneous ICD implantation in primary prevention. The proband is currently undergoing optimized medical therapy and presents with dyspnea on moderate efforts (NYHA class II); the transthoracic echocardiogram highlights an improvement in the left ventricular systolic function (LVEF 45%, LV GLS −17.5%), and the EKG shows sinus bradycardia and negative T waves from V1 to V3.

### 2.2. Genetic Test

WES of the index case (II:2) was performed and identified a large number of variants, approximately 12,000. When data analysis was restricted to the selected genes associated with DCM (n = 81), a total of 194 variants (12 insertions/deletions, 1 multi-nucleotide variant (MNV), and 181 substitutions) were found. Because the index patient was the only affected family member, we focused our filtering also on de novo occurring gene variants. Selected rare variants were filtered and classified according to the standards and guidelines for the interpretation of human sequence variants of the American College of Medical Genetics and Genomics (ACMG), using Expert Variant Interpreter [20] (Appendix A). The NGS screening did not find the presence of any pathogenic or potentially pathogenic variants in the 81 analysed genes. However, a variant of uncertain significance (VUS) was identified. A germline heterozygous A>G transversion at nucleotide 10103 in the DMD gene (NM_004006.2 Homo sapiens dystrophin (DMD), transcript variant Dp427m, mRNA), leading to an aspartic acid–glycine substitution at amino acid 3368 of the DMD protein (c.10103A>G p.Asp3368Gly, NP_003997.2, dystrophin isoform Dp427m) was identified by the NGS analysis, and confirmed by the PCR-based sequencing of exon 70 according to the Sanger method. The DMD gene is associated with an X-linked form of dilated cardiomyopathy type 3B (Phenotype MIM number 302045). Currently, the c.10103A>G (p.Asp3368Gly) variant is not reported in databases of human gene mutations [GnomAD, ClinVar, or in the National Center for Biotechnology Information-Single Nucleotide Polymorphism (NCBI SNP) database], and the data about its frequency in humans are not available; thus, we classified it as a novel rare variant.

### 2.3. Segregation Analysis

After the identification of the *DMD* mutation in the index case, clinical evaluation and genetic screening for this specific variant were extended to the first-degree relatives of the proband (I-1, I-2, II-3, III-1). The proband’s father, mother, and sister underwent clinical evaluation, EKG, and echocardiography and were found to be phenotypically healthy. The young boy did not present any clinical and echocardiographic signs or symptoms attributable to DCM. On the EKG, negative T waves from V1 to V4, compatible with the age, were evident (Figure 2).

However, he reported asthenia and bilateral calf hypertrophy at clinical examination was found. Laboratory investigations revealed increased levels of creatinine kinase (values ranging from 3600 to 19.000 units/L, n.v. < 170 units/L). Moreover, the young patient is suffering from obstructive sleep apnea syndrome (OSAS) and irritable bowel syndrome (IBS) with a tendency for constipation. In both parents, direct sequencing of the *DMD* exon 70 did not reveal the occurrence of the gene variant (Figure 3A). Thus, the p.Asp3368Gly substitution was considered as occurring de novo. Subsequently, the gene variant was investigated in the proband’s 8-year-old son, who was identified as hemizygous for the same variant. Following results from the laboratory and genetic analyses, he was referred to an out-patient clinic for degenerative and muscle diseases, where the diagnosis of “unspecified myopathy due to DMD defect” was made. The patient is now undergoing serial clinical evaluation.

### 2.4. In Silico Analysis

Effects of the p.Asp3368Gly substitution were explored by using in silico pathogenicity prediction tools. The NCBI GenBank accession number NM_004006.3 and UniProt identifier P11532 were used as the reference sequences. PolyPhen-2, SIFT, REVEL, MetaL, and combined annotation-dependent depletion (CADD) scores were calculated to evaluate the pathogenicity of the identified variant. In silico predictions suggested that this variant could have a deleterious impact on protein structure and function (Table 1).

According to the American College of Medical Genetics (ACMG) criteria for the classification of human gene variants, the p.Asp3368Gly was identified as PM2 (absent from controls in GnomAD, Exome Sequencing project, 1000 Genomes, or ExAC; accessed on 18 December 2023) and PP3 (multiple computational programs predictive of the impact of a gene mutation on the structure and/or function of the encoded protein supported a possible deleterious effect of the T point variant c.10103A>G on DMD protein). Thus, p.Asp3368Gly was classified as a rare clinical VUS [20]. The comparison of the human DMD with orthologues sequences revealed a high conservation of aspartic acid at codon 3368 of the DMD protein among different species. The identified variant p.Asp3368Gly causes a protein change from a highly conserved aspartic acid across species (Figure 3B).

### 2.5. Limitations

The present study has some limitations because longitudinal functional data are lacking. However, genotype–phenotype segregation within the family, absence in population databases (GnomAD exome and ExAC), finding a different additional variation in the same codon of a DMD/BMD patient, and the in silico prediction of a deleterious effect support the causative role of the DMD p.Asp3368Gly heterozygous variant. The present study evaluated only a single family and results cannot be generalized.

## 3. Discussion

We report a de novo gene variant mutation (p.Asp3368Gly) in the *DMD* gene associated with dilated cardiomyopathy occurring at the age of 33 years in a young woman, and with peripheral myopathy in her 8-year-old son. The NGS-based genetic screening of a set of genes involved in DCM identified a novel germline heterozygous missense variation in the *DMD* gene. The index case carried the de novo p.Asp3368Gly variant in heterozygosity and was affected by DCM with biventricular involvement and a mildly reduced left ventricular ejection fraction. Her 8-year-old son, carrying the same variant in hemizygosity, was diagnosed with unspecified myopathy and until now did not manifest any signs or symptoms of cardiac involvement, but showed clinical signs of peripheral myopathy with bilateral calf hypertrophy and highly increased CK values.

XLDCM may phenotypically overlap with DCM due to other causes, and there is increasing awareness that disease manifestations may not be restricted to young males. Although XLDCM has been known for a long time, knowledge of the genetic determinants of this disorder is incomplete since *DMD* variants have been identified in a minority of males with suspected XLDCM [15,20].

Mutations in the *DMD* gene have been associated with five allelic phenotypic variants: Duchenne MD (310200); its milder variant, Becker MD (300376); the more rarely occurring XLDCM (302045); exercise-induced myalgia with myoglobinuria; and “hyperCKemia” without clinical symptoms or signs. Women who carry a heterozygous *DMD* variant are at increased risk for cardiomyopathy and may have mild skeletal muscle symptoms [21]. The large majority of gene variants are deletions (60%), whereas other variants are duplications (7%) of one or more exons, small insertions or deletions within an exon (7%), or single-nucleotide point mutations being identified only in a minority of patients (20%) [22]. Interestingly, a different variation in the same codon (p.Asp3368Tyr) was reported in a man with a clinical diagnosis of DMD/BMD [23].

The *DMD* gene is located on the human chromosome Xp21, contains 79 exons, and spans more than 2500 kb [24]. The gene encodes for the 427 kDa cytoskeletal protein dystrophin, which represents the full-length dystrophin gene and has three promoters: the M promoter produces the Dp427m isoform, expressed in skeletal and cardiac muscle; the B promoter produces Dp427c, expressed in the brain; and the P promoter produces Dp427p, expressed in the Purkinje cells in the brain [25]. An additional four internal promoters give rise to shorter dystrophin isoforms (Dp260, Dp140, Dp116, and Dp71) with specific tissue expression [25]. Protein structural domains include two calponin-homology domains (CH1 and CH2), four hinges (H1 to H4), a central rod composed of twenty-four spectrin repeats (R1 to R24), a cysteine-rich domain (CRD) encompassing a WW domain, two EF-hands, a ZZ-type zinc finger domain, and the carboxy-terminal domain (CTD) [16]. The structure of dystrophin and its stability may correlate with the pathogenesis of cardiac involvement. Patients diagnosed with XLDCM or mild BMD with DCM have various types of gene variants such as missense, nonsense, deletion, insertion, or duplication of the *DMD* gene, spanning from exon 1 to exon 63. Although uncommon, variations located between exon 63 and exon 79 cause loss of all the dystrophin isoforms and have been associated with XLDCM or mild BMD with DCM [26].

Patients with DCM associated with dystrophin gene variation without severe skeletal myopathy have a high rate of major adverse cardiac events (about 22%) at long-term follow-up appointments, and a portion of them (about 18%) can develop end-stage heart failure. Men can be affected earlier, and a decrease in LVEF at baseline was associated with major adverse cardiovascular events [15,27]. Recently, this subgroup of DCM has been shown to carry a higher risk of end-stage HF but a lower risk of life-threatening arrhythmias [20]. Therefore, a multidisciplinary approach including imaging and genetic testing is needed [28].

While genetic variants causing the disease are well-known, no curative therapy has been developed to date. The development of genome-editing technologies provides new opportunities to correct a variant responsible for DMD. Correction approaches for DMD include permanent exon removal, exon skipping, exon reframing, and exon knock-in. Functional dystrophin gene restoration has been demonstrated by CRISPR/Cas9 editing in iPSCs derived from DMD patients with exon deletions, exon duplications, and point mutations [29]. Further studies are required to develop therapeutic strategies to correct the detrimental effect of pathogenic *DMD* variants. 

## 4. Conclusions

We report an early presentation of dilated cardiomyopathy in a 33-year-old woman due to a de novo *DMD* gene variant (p.Asp3368Gly). Genetic identification of this mutation allowed an early diagnosis of skeletal muscle disease in her son.

## Figures and Tables

**Figure 1 ijms-25-02787-f001:**
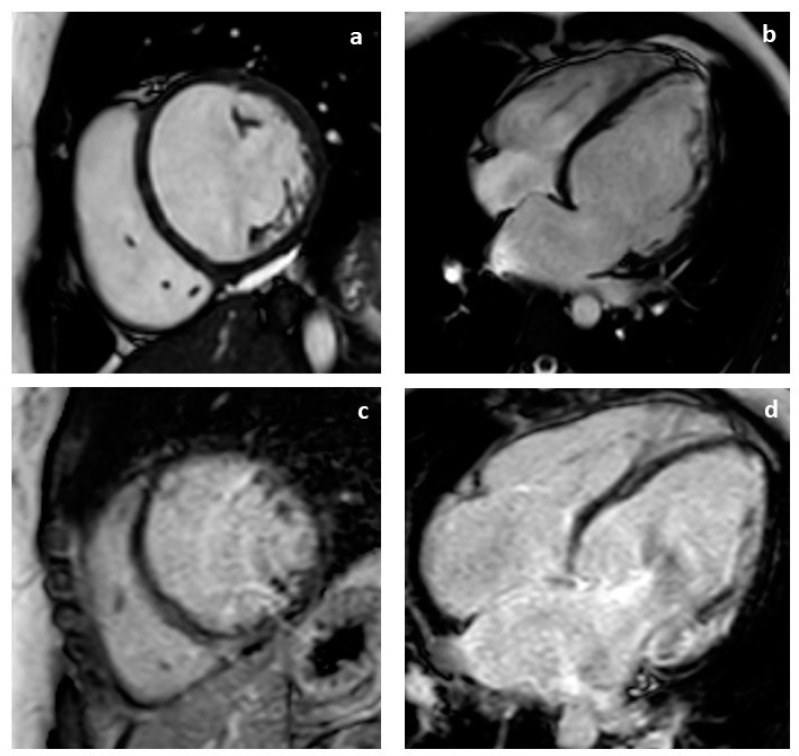
Cardiac magnetic resonance of the proband (33-year-old woman) with mutation of the *DMD* gene. Short axis (**a**) and four chamber view (**b**) Cine-Balanced-TFE sequences showing severe dilatation of the left atrium and left ventricle. Short axis (**c**) and four chamber view (**d**) Cine-Balanced-TFE sequences showing late gadolinium enhancement (LGE) in the mesocardial inferior interventricular septum, infero-lateral wall of the left ventricle and free wall of the right ventricle.

**Figure 2 ijms-25-02787-f002:**
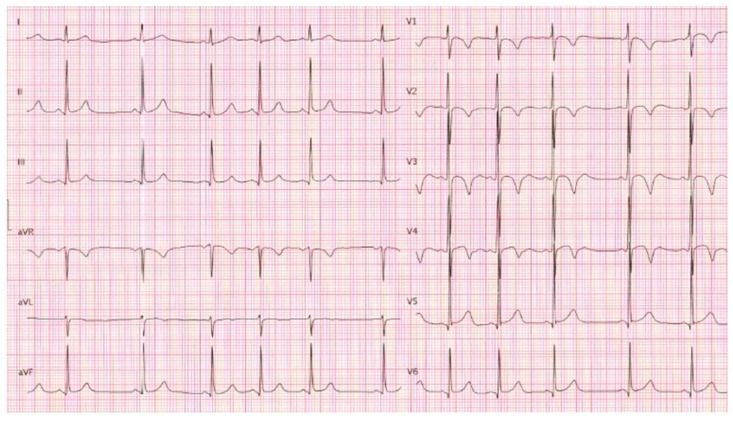
A 12-lead electrocardiogram of the 8-year-old son of the proband.

**Figure 3 ijms-25-02787-f003:**
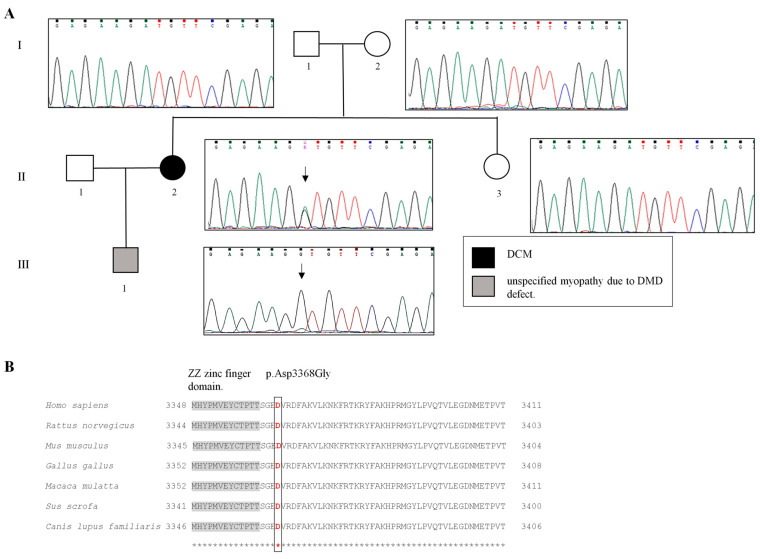
(**A**). Pedigree of the proband carrying the DMD variant. In the pedigree, symbols for each generation in the family are placed in a row and numbered with Roman numerals. Within each generation, people are numbered with Arabic numerals. Circles indicate females, squares males. Open symbols represent asymptomatic individuals; Black symbol indicate propositus affected by DMC and grey one by unspecified myopathy. Sanger sequencing showing the c.10103A>G substitution in the propositus and her son (black arrows) (**B**). Alignment of a domain and the amino acid position number 3368 (highlighted) along different species. UniProt identifier P11532 was used as the human reference protein sequence.

**Table 1 ijms-25-02787-t001:** In silico predicted pathogenicity of the *DMD* variant.

Gene	Nucleotide	AA Change	SIFT	Polyphen	CADD	REVEL	Metal R
* **DMD** *	G**A**T/G**G**T	D3368G	Deleterious	Probably damaging	**30**	Damaging	Damaging
			score 0.01	score 0.975		score 0.874	score 0.844

## Data Availability

The data presented in this study are available on request from the corresponding author. The data are not publicly available due to privacy and ethical restrictions.

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
