# Peer review of "De Novo p.Asp3368Gly Variant of Dystrophin Gene Associated with X-Linked Dilated Cardiomyopathy and Skeletal Myopathy: Clinical Features and In Silico Analysis"

_ijms, 2024, doi:10.3390/ijms25052787_

Round 1

Reviewer 1 Report

Comments and Suggestions for Authors

The article “De Novo p.Asp3368Gly Mutation of Dystrofin Gene Associated with X-Linked Dilated Cardiomyopathy and Skeletal Myopathy: Clinical Features and In Silico Analysis” is interesting for readers and is devoted to the important problem of the connection between point mutations and serious diseases such as X-Linked Dilated Cardiomyopathy and Skeletal Myopathy. This paper is recommended for publication after minor revision.

Lines 25-26. The genetic screening for this specific variant was extended to the first-degree 25

relatives of the proband (his mother, father her sister) and due to absence of the variant in both 26 parents, p.Asp3368Gly is considered de novo.

Please specify which relatives were studied. The statement in parentheses is not clear.

2.5 & 3.4 In silico analysis

Too little detail about in silico analysis is provided. Please add input information such as protein ID so that other researchers can perform the same analysis.

Figure 3

A: The figure is blurry. Colors and signatures are indistinguishable.

B: Please indicate the database to which codes P11532, etc. correspond. Add unencrypted species names to the figure caption.

Author Response

Lines 25-26. The genetic screening for this specific variant was extended to the first-degree relatives of the proband (his mother, father her sister) and due to absence of the variant in both 26 parents, p.Asp3368Gly is considered de novo.

Please specify which relatives were studied. The statement in parentheses is not clear.

As suggested, the sentence has been amended and relatives investigated specified.

2.5 & 3.4 In silico analysis

Too little detail about in silico analysis is provided. Please add input information such as protein ID so that other researchers can perform the same analysis.

Information required, such as The NCBI GenBank accession number NM_004006.3 and UniProt identifier P11532, has been added.

Figure 3

A: The figure is blurry. Colors and signatures are indistinguishable.

B: Please indicate the database to which codes P11532, etc. correspond. Add unencrypted species names to the figure caption.

Figure 3 has been modified as reviewer’s suggestions.

Reviewer 2 Report

Comments and Suggestions for Authors

In the original manuscript 'De novo p.Asp3368Gly Mutation of Dystrofin Gene Associated with X-linked Dilated Cardiomyopathy and Skeletal Myopathy: Clinical Features and In Silico Analysis, the authors identified a novel de novo mutation in the DMD gene in a female patient with dilated cardiomyopathy. The topic of this manuscript is interesting. However, I suggest several changes:

1.) All human gene names (like e.g., DMD) should be written in the complete manuscript in Italics.

2.) Several corrections are necessary: Glycine not glicine. Dystophin not dystofin. I suggest that a native English speaker should correct this manuscript.

3.) Several abbreviations (like DCM) are introduced several times. Please change and use after introduction of an abbreviations this in the following text.

4.) Please prepare a list of rare variants (based on the WES) and present this as a table within the supplements.

5.) Line 104/105: Please indicate the sequences of the used primers.

6.) After Line 43: Please add shortly an overview about the genetic background of DCM. The book chapter 'The genetic landscape of cardiomyopathies' is helpful in this  context. I would mention the major DCM gens (TTN, LMNA, RBM20) and DES. Mutations in the DES gene cause similar to DMD also cardiac and skeletal myopathies. Therefore, I would also mention this gene here (see Molecular insights into cardiomyopathies associated with desmin (DES) mutations).

7.) Line 211: Could you please add the GNOM-AD database here and check that this mutation is absent also in this database?

8.) I would indicate the date, when database search was performed (Line 210/211), since the databases can be updated in the future.

9.) Although the authors did not present novel functional data, the authors can discuss novel developments for genetic repair of the DMD gene like exon skipping or genome editing. Several papers of the group of Eric Olson have shown efficient genome editing of DMD. Therefore, I suggest to discuss these novel therapies as an outlook.

In summary, I suggest a major revision and would be happy to re-review this manuscript in the future. Good luck! 

Comments on the Quality of English Language

Moderate changes are necessary. I suggest a correction by a native English speaking editor.

Author Response

  • All human gene names (like e.g., DMD) should be written in the complete manuscript in Italics.

The human gene names are indicated in Italics throughout the present version of the manuscript.

  • Several corrections are necessary: Glycine not glicine. Dystophin not dystofin. I suggest that a native English speaker should correct this manuscript.

As suggest we correct all typos and the text was checked by a native English speaker.

  • Several abbreviations (like DCM) are introduced several times. Please change and use after introduction of an abbreviations this in the following text.

The text has been modified according to the reviewer’s suggestion.

  • Please prepare a list of rare variants (based on the WES) and present this as a table within the supplements.

A list of rare variants found has been reported in the present version of supplementary material.

  • Line 104/105: Please indicate the sequences of the used primers.

The sequences of the used primers have been reported in Materials and Methods section 2.4.

  • After Line 43: Please add shortly an overview about the genetic background of DCM. The book chapter 'The genetic landscape of cardiomyopathies' is helpful in this context. I would mention the major DCM gens (TTN, LMNA, RBM20) and DES. Mutations in the DES gene cause similar to DMD also cardiac and skeletal myopathies. Therefore, I would also mention this gene here (see Molecular insights into cardiomyopathies associated with desmin (DES) mutations).

As the Reviewer suggested a short overview about the genetic background of DCM has been added in the present version of the manuscript.

  • Line 211: Could you please add the GNOM-AD database here and check that this mutation is absent also in this database?

The Gnom-AD database has been added in the in-silico analysis section. Variant is not reported in the Gnom-AD database.

  • I would indicate the date, when database search was performed (Line 210/211), since the databases can be updated in the future.

As the Reviewer suggested the when database search was performed has been added.

  • Although the authors did not present novel functional data, the authors can discuss novel developments for genetic repair of the DMD gene like exon skipping or genome editing. Several papers of the group of Eric Olson have shown efficient genome editing of DMD. Therefore, I suggest to discuss these novel therapies as an outlook.

Thank you for the helpful comment. As the Reviewer suggested we report a short discussion about novel therapeutic strategies such as genome-editing technologies.

Round 2

Reviewer 2 Report

Comments and Suggestions for Authors

The authors have addressed all points, which I have mentioned before. Therefore, I suggest to accept this manuscript for publication.